# The use of artificial intelligence in induced pluripotent stem cell-based technology over 10-year period: A systematic scoping review

Quan Duy Vo[1,2]*, Yukihiro Saito[3], Toshihiro Ida[1], Kazufumi Nakamura[1], Shinsuke Yuasa[1]

1 Faculty of Medicine, Department of Cardiovascular Medicine, Dentistry and Pharmaceutical Sciences, Okayama University, Okayama, Japan, 2 Faculty of Medicine, Nguyen Tat Thanh University, Ho Chi Minh City, Viet Nam, 3 Department of Cardiovascular Medicine, Okayama University Hospital, Okayama, Japan

* dr.duyquan@gmail.com

**Data Availability Statement:** All relevant data are within the manuscript and its Supporting Information files.

## Abstract

### Background

Stem cell research, particularly in the domain of induced pluripotent stem cell (iPSC) technology, has shown significant progress. The integration of artificial intelligence (AI), especially machine learning (ML) and deep learning (DL), has played a pivotal role in refining iPSC classification, monitoring cell functionality, and conducting genetic analysis. These enhancements are broadening the applications of iPSC technology in disease modelling, drug screening, and regenerative medicine. This review aims to explore the role of AI in the advancement of iPSC research.

### Methods

In December 2023, data were collected from three electronic databases (PubMed, Web of Science, and Science Direct) to investigate the application of AI technology in iPSC processing.

### Results

This systematic scoping review encompassed 79 studies that met the inclusion criteria. The number of research studies in this area has increased over time, with the United States emerging as a leading contributor in this field. AI technologies have been diversely applied in iPSC technology, encompassing the classification of cell types, assessment of disease-specific phenotypes in iPSC-derived cells, and the facilitation of drug screening using iPSC. The precision of AI methodologies has improved significantly in recent years, creating a foundation for future advancements in iPSC-based technologies.

### Conclusions

Our review offers insights into the role of AI in regenerative and personalized medicine, highlighting both challenges and opportunities. Although still in its early stages, AI

**Funding:** The author(s) received no specific funding for this work.

**Competing interests:** The authors have declared that no competing interests exist.

**Abbreviations:** AI, artificial intelligence; aiPSC, artificially induced pluripotent stem cell; ALS, amyotrophic lateral sclerosis; AUC, area under the curve; CM, cognitive maps; CNN, convolutional neural network; CT, Classification Tree; DAGSVM, Directed Acyclic Graph; DL, Deep learning; ECC, embryonal carcinoma cell; ESC, embryonic stem cells; hiPSC, Human induced pluripotent stem cells; ICC, intraclass correlation coefficient; IOPD, Infantile onset Pompe disease; iPSC, Induced pluripotent stem cells; k-NN, k-Nearest Neighbour; lncRNA, long non-coding RNA; MEA, microelectrode array; ML, Machine learning; MPI, magnetic particle imaging; NB, naïve Bayes; PRISMA, Preferred Reporting Items for Systematic Reviews and Meta-Analysis; PSC, pluripotent stem cell; RNN, recurrent neural network; ROC, Receiver Operating Characteristic; SVM, Support Vector Machines; wndchrm, compound hierarchy of algorithms representing morphology.

technologies show significant promise in advancing our understanding of disease progression and development, paving the way for future clinical applications.

# Introduction

Stem cell research has advanced significantly since 1961, when bone marrow-derived multipotent stem cells were first identified [1]. Stem cells are unique cells with the capability to continually replicate through mitosis, leading to the formation of more cells. This process generates two distinct cell types: one that evolves into a specific cell type and another that retains the capability of self-renewal [2]. Stem cells are broadly classified into three types: induced pluripotent stem cell (iPSC), embryonic stem cell (ESC), and adult stem cell (ASC) [3]. The iPSC and ESC are categorized as pluripotent stem cell (PSC) due to their capacity to transform into the three germ layers: ectoderm, mesoderm, and endoderm. In 2006, Kazutoshi Takahashi and Shinya Yamanaka successfully transformed mouse somatic cells into iPSC by introducing specific transcription factors known as Oct4, Sox2, Klf4, and c-Myc using a viral vector [4]. Following this, various methods have been used to reprogram different types of mouse and human somatic cells into iPSC [5]. This innovative method of reprogramming human cells has sparked immense interest in both scientific and medical fields. iPSC provides an alternative to human ESC as a source of pluripotent cells. A significant advantage of induced pluripotent stem cells is that they are derived from somatic cells that could be obtained non-invasively. These cells carry the individual's genetic characteristics, which can reduce the risk of immune rejection [6].

The attention to iPSC-based therapies is increasing in the field of modern medicine. Their application in disease modelling, drug screening, and regenerative medicine is expanding exponentially [7]. The iPSC is pivotal in disease modelling due to its self-renewal capability and ability to differentiate into all human body cell types. This makes them ideal for creating various disease models for research [8–10]. Patient-specific iPSC is particularly valuable in developing targeted therapeutic strategies and drug development. Furthermore, iPSC from both normal and diseased cells can be differentiated into neurons, hepatocytes, cardiomyocytes, etc., for evaluating toxicity and side effects, which are critical factors in the development of therapeutic molecules [11]. In regenerative medicine, iPSC is being used to repair or regenerate damaged or degenerated tissues. This is achieved by creating the organ tissues in laboratories from iPSC and transplanting them to the injured area. This therapy holds promise for treating conditions such as hematopoietic disorders, musculoskeletal injuries, spinal cord injuries, and liver damage [12–14].

Various techniques have been developed for creating iPSC, such as using retroviruses or lentiviruses for gene transduction and chemical induction. However, the process of generating iPSC is typically slow and not very efficient, taking about 1–2 weeks for rodent cells and 3–4 weeks for human cells, with generally low success rates. Moreover, evaluating the quality of iPSC by examining colony morphology is prone to human error, presenting a significant challenge that must be addressed before pursuing further experimental or therapeutic uses. Despite the advancements in enhancing both the efficiency and speed of iPSC cultivation, this process remains resource-consuming, necessitating the development of an automated system to minimize errors and enhance iPSC analysis. Recently, artificial intelligence (AI) technologies, including machine learning (ML) and deep learning (DL), have been employed to enhance regenerative therapy. The implementation of these AI-driven approaches could refine the

management of clinical trials for innovative stem cell therapies designed for a multitude of diseases. AI contributes to the personalization of treatment protocols for individual patients, the forecasting of clinical outcomes, and the practical organization of patient recruitment. These advancements have the potential to diminish the challenges inherent in these trials and reduce the overall expenses [15].

AI technology involves the development of computer systems designed to perform tasks traditionally requiring human intelligence, encompassing learning, reasoning, perception, and problem-solving. The objective of AI systems is to mimic human cognitive abilities, functioning autonomously and continuously improving their capabilities by learning from data and past experiences [16]. The concept of AI has existed for many years, but recent advancements in ML and DL have enabled the development of more complex AI systems. Nowadays, AI systems can process vast datasets to predict outcomes, categorize objects, and execute other intricate functions.

Machine learning represents a sophisticated and advancing technology that enables computers to identify and categorize patterns in extensive datasets without explicit programming. It is an interdisciplinary field, intertwining computer science, mathematics, philosophy, control theory, determinism, and other areas [17]. This approach focuses on replicating or emulating human learning processes using computers. These techniques allow researchers to filter through extensive datasets, identify patterns, make predictions, learn from errors, and adapt their methods for improvement without explicit programming [18].

Deep learning is a specialized branch of ML which employs artificial neural networks, and some other algorithms, for data processing. These networks are structured to resemble the human brain, enabling the identification of more intricate patterns and decision-making based on trained data [19]. This method has significantly transformed the AI landscape, equipping machines to undertake tasks previously considered unachievable. A notable strength of DL lies in its capacity to manage extensive and complex datasets. Deep learning algorithms can process millions of data points and discern patterns beyond human recognition [20]. Furthermore, DL distinguishes itself with its adaptive learning capabilities over time, whereas traditional ML methods often lack a memory component and require manual tuning [21,22]. This self-optimization allows DL algorithms to progressively enhance their performance as they are exposed to more data. Among these algorithms, the convolutional neural network (CNN) is predominantly utilized for image classification tasks [23].

AI has been instrumental in advancing iPSC technology, especially through non-invasive cell identification, genomic and proteomic data analysis, and the enhancement of targeted therapies [24] (Fig 1). The ongoing advancements in AI hold a promise to radically transform medical science. However, given the rapid pace of this field, it is imperative to conduct extensive research and validation to leverage the full potential of AI in iPSC technology. This review aims to investigate and illustrate a comprehensive review of existing studies on how AI-based methodologies contribute to the advancement of iPSC technology.

## Materials and methods

This systematic review was conducted adhering to the 2020 Preferred Reporting Items for Systematic Reviews and Meta-Analysis (PRISMA) guidelines and principles [25].

### Searching strategy

In December 2023, a primary search was conducted across three electronic databases: PubMed, Web of Science, and Science Direct. The search utilized terms and their synonyms, including "artificial intelligence," "deep learning," "machine learning," and "induced

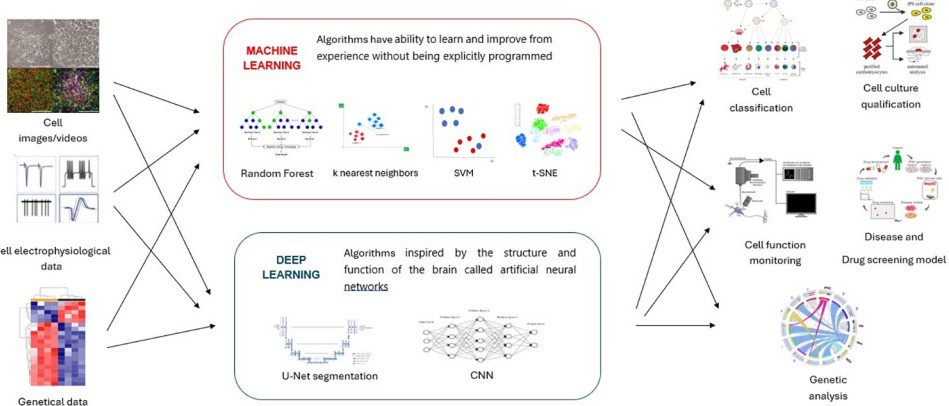

**Fig 1. The application of AI technology in the iPSC field.**

pluripotent stem cell," to identify potential articles in the targeted databases using the specified search terms. After this initial step, duplicate entries were removed. The next step involved a preliminary screening of the remaining papers by reviewing their titles and abstracts. This was followed by a thorough full-text review of each paper, conducted independently by two investigators to determine their relevance and eligibility. A third investigator was brought in to provide a definitive judgment if there were any disagreements or uncertainties.

Additionally, to ensure comprehensive coverage, a manual citation search was also undertaken. This involved screening the reference lists of all retrieved studies to uncover any other pertinent research that might have been missed in the initial database search.

## Inclusion and exclusion criteria

This study included quantitative research that applied AI-based imaging analysis to iPSC from either animals or humans. No language or publication date limitations were imposed, given the emerging nature of this field. Exclusions were made for preprints, conference papers, qualitative studies, and quantitative studies focusing on AI technology for cell types other than iPSC. Studies were also excluded if the full text was not available.

## Quality assessment

All studies meeting the inclusion criteria were evaluated for the best practice in AI research according to DOME guideline [26].

## Data extraction

Data from the included studies were independently extracted and assessed by two reviewers. If there were any disagreements, a third reviewer would be consulted. Quality assessment techniques were applied to ensure that studies met the inclusion criteria. Extracted data encompassed bibliographic details (authors, publication year, funding, title, language, journal), and details of the iPSC line, AI technique, and the accuracy of the AI algorithm were also collected.

## Results

### Search result

In general, from the initial 872 articles retrieved from three electronic databases, 440 were removed after deduplication. Following this, 432 studies underwent screening of titles and

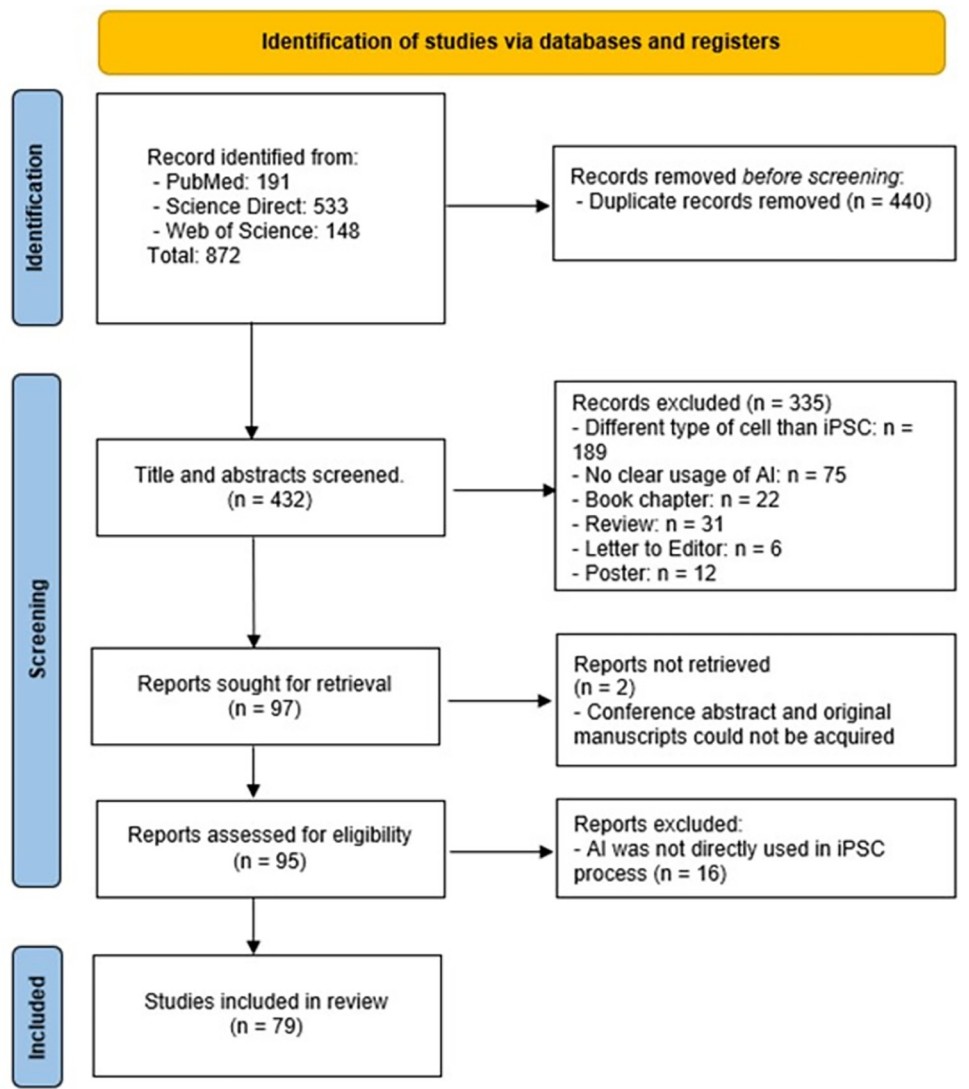

**Fig 2. Study flow-chart.**

abstracts based on the inclusion criteria. After this initial screening, 95 articles were selected for full-text analysis. However, 16 studies were excluded after a detailed review, resulting in 79 studies being included in this systematic review (Fig 2).

## Characteristics of included studies

Of the 79 studies, one focused on animal iPSC, 75 on human iPSC, and three utilized artificially created iPSC. Additional details about these studies can be found in S1 Table in S1 File.

The demographic analysis highlighted the United States as the predominant contributor in this field, accounting for 31 of the 79 studies (39.2%). Europe and Japan followed, each contributing 15 studies (18.9%). Other regions, including China, the United Kingdom, Taiwan, Korea, and Canada, had a lower research volume, with contributions ranging from 2 to 4 studies per region. Despite year-to-year variance, there is an increasing trend of publication volume regarding the application of AI in the iPSC field over the years, peaking in 2022 with twenty studies published (Fig 3).

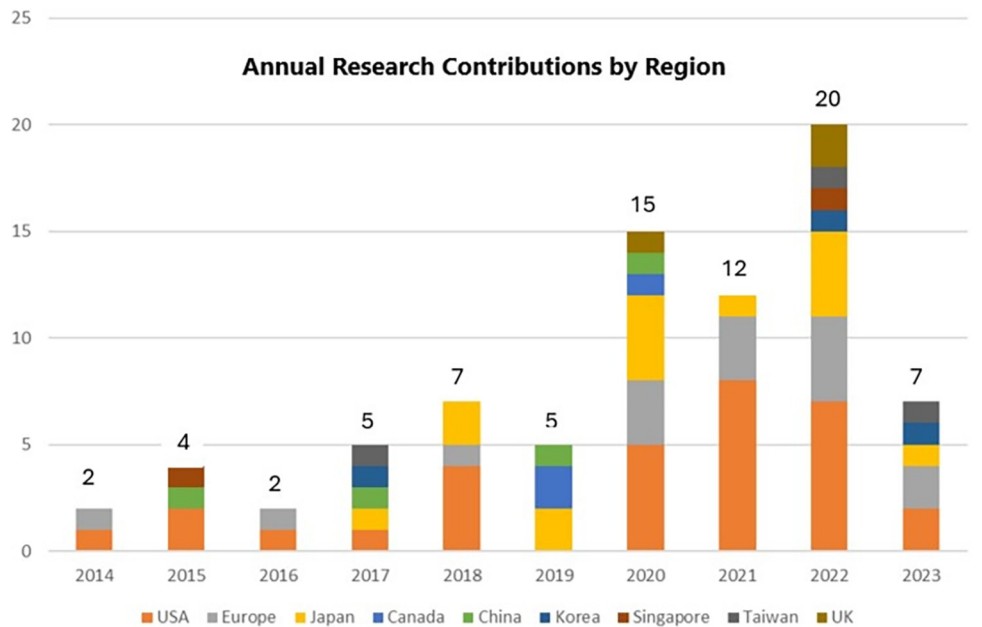

**Fig 3. Number of articles on AI application within iPSC field.**

## Applications of AI in iPSC

The applications of AI in iPSC field can be classified into 5 major categories (Fig 4).

### Cell culturing and processing

A total of 44 articles were identified that leveraged AI technologies to enhance cell processing methods. Among these, 23 studies employed AI for the classification of various cell types based on morphological images, 11 aimed at assessing the biological attributes of cells, five focused on managing the quality of the cell culture process, two utilized AI for monitoring cells after transplantation, and three developed artificial iPSC using existing datasets. Data inputs varied across these studies, with 39 employing cell imagery, two using biological data, and three relying on genetic data (Table 1).

### Genetic analysis

In genetic analysis, ten articles employed AI technologies. Of these, seven studies applied AI for RNA sequencing analysis, two for DNA analysis, and one for evaluating epigenomic characteristics (Table 2).

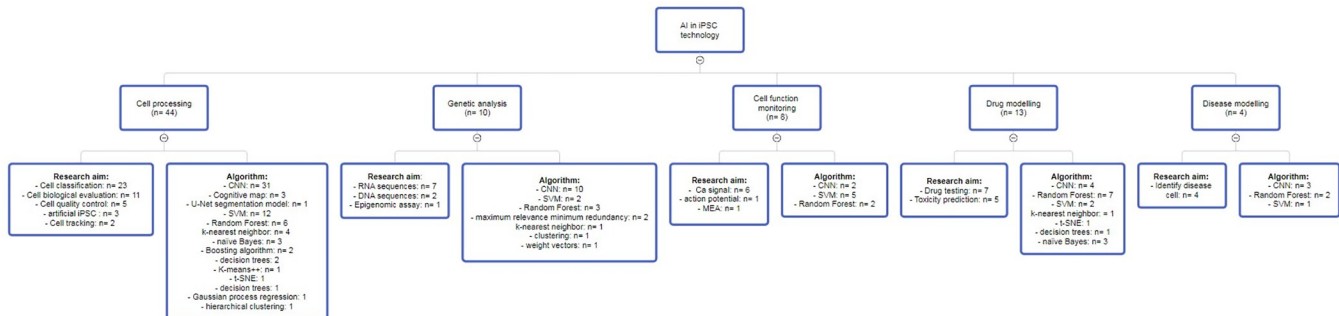

**Fig 4. Major applications of artificial intelligence in iPSC technology.**

**Table 1. The application of AI technology in cell processing.**

| No | Study | Cell type | Input data | Study analysis | Algorithm |
|---|---|---|---|---|---|
| 1 | Aixia S [27] | islet organoid | image | Cell tracking | K-means++ |
| 2 | Anna S [28] | iPS-CM | image | Cell biological evaluation | CNN |
| 3 | Aurore V [29] | iPSC-neuron | image | Cell morphological classification | SVM, linear discriminant analysis |
| 4 | Bianca W [30] | iPS-CM | Biological information | Cell classification | random forest, Gaussian process regression, multivariate adaptive regression spline |
| 5 | Bo J [31] | neural progenitors | image | Cell morphological classification | SVM |
| 6 | Brodie F [32] | iPSC | image | Cell morphological classification | CNN |
| 7 | Cesar A. P [33] | iPSC | image | Cell morphological classification | CNN |
| 8 | Cesar A. P [34] | iPSC | image | Cell morphological classification | CNN |
| 9 | Chia-Chen H [35] | neuron | image | Cell morphological classification | SVM, random forest, k-nearest neighbor |
| 10 | Chung-Yueh L [36] | retinal | Biological information | Cell classification | CNN |
| 11 | Colombine V [37] | neuron | image | Cell morphological classification | CNN |
| 12 | Dai K [38] | iPS-endothelial cells | image | Cell morphological classification | CNN |
| 13 | David A. J [39] | iPSC | image | Cell biological evaluation | CNN |
| 14 | Diogo T [40] | iPS-CM | image | Cell morphological classification | k-nearest neighbor, decision trees, t-SNE, Naïve Bayes classifiers |
| 15 | Jianying G [41] | iPSC | image | Cell biological evaluation | PhenoLOGIC machine learning |
| 16 | Haishan Z [42] | cardiac progenitor | image | Cell morphological classification | Boosting algorithm |
| 17 | Henry J [43] | iPSC | image | Cell biological evaluation | SVM, k-nearest neighbour, naïve Bayes, classification tree |
| 18 | Henry J [44] | iPSC | image | Cell morphological classification | SVM, k-nearest neighbor |
| 19 | Kaivalya M [45] | iPSC | image | Cell morphological classification | random forest, k-nearest neighbor, naïve Bayes, simple logistic |
| 20 | Kazuaki T [46] | iPSC | image | Cell morphological classification | weighted neighbour distances |
| 21 | Ke F [47] | iPSC | image | Cell biological evaluation | CNN |
| 22 | Ke Y [48] | iPS-retinal cell | image | Cell quality control | hierarchical clustering |
| 23 | Ken O [49] | iPS-CM | image | Cell quality control | SVM |
| 24 | Ken O [50] | iPS-CM | image | Cell quality control | CNN |
| 25 | Keonhyeok P [51] | kidney | image | Cell morphological classification | CNN |
| 26 | Kimerly A.P [52] | iPSC | image | Cell quality control | U-Net segmentation model |
| 27 | Letao Y [53] | Hippocampus organoid | image | Cell biological evaluation | Gaussian process regression |
| 28 | Mahnaz M [54] | cardiomyocytes | image | Cell morphological classification | SVM |
| 29 | Minjae K [55] | iPSC | image | Cell biological evaluation | CNN |
| 30 | M. S Kavitha [56] | iPSC | image | Cell morphological classification | SVM, random forest, decision tree, adaptive boosting classifier, Multilayer perceptron classifier |
| 31 | Na Ta [57] | iPSC | Genetic data | Cell tracking | CNN, SVM |
| 32 | Prithvijit M [58] | iPSC | Genetic data | Cell biological evaluation | CNN |
| 33 | Quinton S [59] | iPSC | image | Cell classification | SVM |

*(Continued)*

**Table 1.** (Continued)

| No | Study | Cell type | Input data | Study analysis | Algorithm |
|----|-------|-----------|------------|----------------|-----------|
| 34 | Sally W [60] | iPS-skeletal muscle cell | Genetic data | artificial iPSC | CNN, cognitive maps |
| 35 | Sally W [61] | lung | Genetic data | artificial iPSC | CNN, cognitive maps |
| 36 | Saori A [62] | iPSC | image | Cell biological evaluation | CNN |
| 37 | Scott A [63] | iPSC | image | Cell morphological classification | CNN |
| 38 | Slo-Li C [64] | iPSC | image | Cell morphological classification | CNN |
| 39 | Takashi W [65] | iPSC | image | Cell morphological classification | SVM |
| 40 | Tobias P [66] | iPSC | image | Cell morphological classification | stepbased Stochastic Gradient Descent |
| 41 | Toshiro I [67] | retinal | image | Cell biological evaluation | CNN |
| 42 | Yohei H [68] | iPSC | image | Cell quality control | CNN |
| 43 | Yuan-Hsiang C [69] | iPSC | image | Cell morphological classification | CNN |
| 44 | Wayne R. D [70] | iPSC | Genetic data | artificial iPSC | CNN, cognitive maps |

## Cell function monitoring

Eight studies applied AI for monitoring cell functions, employing various inputs: six studies analysed calcium transient signals, one study focused on action potential signals, and one used microelectrode array (MEA) data (Table 3).

## Disease model

Four studies implemented AI to distinguish diseased cells from healthy controls. This included one study that analysed images of Huntington iPSC to identify diseased colonies based on cell morphology [89]. one study that evaluated iPSC-derived motor neurons from individuals with and without amyotrophic lateral sclerosis (ALS), [90] and two studies that assessed contracting signals to detect abnormal cardiomyocytes [91,92] (Table 4).

## Drug screening model

Thirteen studies applied AI in the context of drug modelling, with eight focusing on identifying the effects of pharmaceuticals and the mechanisms of action of drugs, and the remaining five studies assessed drug-induced toxicity on iPSC (Table 5).

**Table 2. The application of AI technology in genetic analysis.**

| No | Study | Cell type | Study analysis | Algorithm |
|----|-------|-----------|----------------|-----------|
| 1 | Alberto C [71] | iPC-neuron | DNA sequences | CCN |
| 2 | Asato S [72] | iPC-neuron | RNA sequences | maximum relevance minimum redundancy |
| 3 | Boqiao L [73] | iPC-neuron | Epigenomic assay | CNN |
| 4 | C Bardy [74] | iPC-neuron | RNA sequences | randomized forest |
| 5 | Christina V. T [75] | iPSC | RNA sequences | k-nearest neighbor |
| 6 | Koichiro N [76] | iPSC | DNA sequences | SVM |
| 7 | Quan H. N [77] | iPSC | RNA sequences | clustering |
| 8 | S John Liu [78] | iPSC | RNA sequences | SVM, random forest |
| 9 | Thong B N [79] | iPSC | RNA sequences | SVM, maximum relevance minimum redundancy |
| 10 | Wei F [80] | iPSC-CM | RNA sequences | random forest |

**Table 3. The application of AI technology in cell function monitoring.**

| No | Study | Cell type | Input | Study analysis | Algorithm |
|----|-------|-----------|-------|----------------|-----------|
| 1 | Henry J [81] | iPS-CM | Cell signal | Ca transient signal | SVM |
| 2 | Hongbin Y [82] | iPS-CM | Cell signal | Ca transient signal | random forest |
| 3 | Hyun H [83] | iPS-CM | Cell signal | Ca transient signal | SVM |
| 4 | Jeremy K.S. P [84] | iPS-CM | Cell signal | Ca transient signal | SVM |
| 5 | Martti J [85] | iPS-CM | Cell signal | Ca transient signal | SVM, random forest |
| 6 | Nicholas J. S [86] | iPS-retinal cell | image | action potential signal | CNN |
| 7 | Parya A [87] | iPS-CM | Cell signal | Ca transient signal | CNN |
| 8 | Utkarsh T [88] | iPS-neuron | Cell signal | MEA | SVM, random forest |

## Main AI algorithms employed in iPSC research

Regarding the application of AI algorithms, convolutional neural network was used in 33 research studies, comprising 41% of the total analysis. Machine learning methodologies were adopted in 44 studies, making up 55.6% of the comprehensive research effort. Among these methodologies, the most frequently used techniques included support vector machine (SVM), which was featured in 21 studies (26.5%), and random forest model, which was employed in 16 studies (20.2%). Other algorithms such as K-Nearest Neighbor, Cognitive Map,. . . were reported in fewer than 10% of the studies.

Additional information of included studies is provided in S1 Table in S1 File.

## Quality of included studies

Of the 79 studies, data and computer codes were accessible in 29% of the cases, and only two studies provided best practice reports [29,97]. However, 85% of the studies provided detailed descriptions of the employed AI methodologies, while 87% reported the performance metrics. Furthermore, the practices of data splitting and hyperparameter tuning were reported in 77% and 67% of the studies, respectively. Detailed information can be found in S2 Table in S1 File.

## Discussion

Over the past twenty years, induced pluripotent stem cell (iPSC) technology has undergone significant advancements and is now being explored for its potential in regenerative medicine. The iPSC therapy holds the promise of creating individualized treatments, avoiding the ethical concerns associated with embryonic stem cells and the risk of immune rejection. However, the adaptation of iPSC technology for clinical applications is facing significant hurdles, including low efficiency and considerable variability in iPSC reprogramming and differentiation methods [106,107], as well as the emergence of undifferentiated and teratoma-induced phenotypes

**Table 4. The application of AI technology in disease modelling.**

| No | Study | Cell type | Disease type | Input | Study analysis | Algorithm |
|----|-------|-----------|--------------|-------|----------------|-----------|
| 1 | Adam W [89] | iPS-neuron | Huntington | image | Identify disease cell | CNN |
| 2 | Keiko I [90] | iPS-neuron | ALS | image | Identify disease cell | CNN |
| 3 | Martti J [91] | iPS-CM | dilated cardiomyopathy and LQT 2 | Cell signal | Identify disease cell | SVM, random forest |
| 4 | Martti J [92] | iPS-CM | dilated cardiomyopathy, LQT 1 and 2, Brugada syndrome, ventricular arrhythmia | Cell signal | Identify disease cell | CNN, random forest |

**Table 5. The application of AI technology in drug modelling.**

| No | Study | Cell type | Input | Study analysis | Algorithm |
|---|---|---|---|---|---|
| 1 | Andrew K [93] | iPS-CM | Cell signal | Drug testing | CNN, SVM, random forests, t-SNE, |
| 2 | Anna S. M [94] | Mid brain organoid | image | Toxicity prediction | random forest |
| 3 | Christopher H [95] | iPS-CM | Cell signal | Drug testing | decision trees |
| 4 | Eugene K. L [96] | iPS-CM | image | Drug testing | SVM |
| 5 | Francis G [97] | iPS-CM | Cell signal | Toxicity prediction | CNN |
| 6 | Hongbin Y [98] | iPS-CM | Cell signal | Drug testing | random forest |
| 7 | Karthikeyan K [99] | iPS-renal cell | Biological information | Toxicity prediction | random forest |
| 8 | Manuela J [100] | embroid | image | Toxicity prediction | random forest |
| 9 | Mahnaz M [101] | iPS-CM and hepatocyte | image | Toxicity prediction | CNN |
| 10 | Martti J [102] | iPS-CM | Cell signal | Drug testing | naïve Baye, random forests |
| 11 | N. Matsuda [103] | iPS-neuron | image | Drug testing | CNN |
| 12 | Tadashi H [104] | iPS-neuron | Biological information | Drug testing | SVM, k-nearest neighbor, random forest |
| 13 | Yuki H [105] | iPS-neuron | image | Drug testing | random forest |

[108]. The integration of artificial intelligence (AI) has significantly advanced the field of iPSC research. This advancement is evident in various aspects, including the classification of iPSC colonies, the non-invasive differentiation between normal and abnormal cells, and the analysis of cellular morphology. Furthermore, AI has been implemented in other areas of iPSC-based technology, encompassing drug testing, disease modelling, and regenerative treatments [109].

## The evolution of AI applications in iPSC research

The evolution of AI applications in iPSC technology over the last decade highlights a transformative journey from simplicity to complexity, both in terms of data and analytical methods. In the early stages, AI is predominantly used to analyse cell image datasets [31,38,43]. These initial studies laid the groundwork for using machine learning to automate tasks like cell classification and tracking, employing relatively straightforward algorithms such as support vector machine (SVM) [44,54], and random forest [56]. These algorithms were adept at parsing visual data, aiding researchers in achieving a significant understanding of cell morphology and behaviour through images.

In the early 2020s, iPSC research witnessed a shift towards incorporating more complex and varied types of data, including genetic information [60] and biological data [30,104]. This transition was facilitated by the broader availability of high-throughput sequencing technologies and advanced imaging techniques, which produced richer datasets capturing the vast complexities of cellular biology. The applications of AI expanded in scope and ambition, moving from basic image recognition tasks to more sophisticated analyses like the modelling of disease processes and the examination of iPSC clonal expansion. This period also saw an increased integration of advanced machine learning algorithms, particularly deep learning models such as convolutional neural network (CNN), which were able to handle the intricacies of large and complex datasets.

The integration of AI technology in iPSC research not only aimed at enhancing the understanding of cellular processes but also at tackling emerging challenges in regenerative medicine and disease modelling [24]. The advancements in computational power and algorithmic sophistication allowed researchers to delve deeper into the biological intricacies of iPSC, paving the way for groundbreaking applications in disease modelling Witmer [90,92] and drug discovery [97,103]. This evolution from simpler machine learning techniques to advanced deep learning approaches mirrors the growing complexity and scale of biological data,

highlighting a trend towards increasingly ambitious and nuanced scientific inquiries in the field of iPSC technology.

## Current applications of AI in iPSC research

Within the field of induced pluripotent stem cells (iPSC), a significant challenge arises from genomic instability initiated by the reprogramming process [110] and variations in the culture system [111]. These factors can affect the differentiation capabilities of iPSC, consequently influencing their therapeutic potential. Also, iPSC derivatives sometimes remain in the fetal stage of development, which may reduce the efficacy of iPSC-based treatments. To ensure the safety and effectiveness of iPSC therapy, it is crucial to continuously evaluate the cells derived from iPSC at different stages of their development. Currently, this assessment mainly depends on the judgment of experienced cell culture specialists. These experts often assess iPSC development and maturity by examining changes in cell shape and the expression of specific cellular markers. This method is not only time-consuming but also susceptible to personal bias. Relying solely on manual techniques for cell quality evaluation is not feasible for mass-producing therapeutic cells [112]. AI technology has consistently demonstrated its effectiveness in assisting the cultivation and maintenance of iPSC, particularly in identifying cell colonies and their functions. Since the pioneering work by Henry Joutsijoki et al. in 2014, which used ML to evaluate the quality and categorize iPSC colonies [43], many studies have applied a wide range of AI algorithms for analysing differentiation levels and morphological alterations within iPSC cultures. With the enhancements in accuracy, AI algorithms have established a reliable framework for various applications in iPSC research, including the accurate classification of iPSC colonies, identification of cellular morphology, non-invasive characterization of cell therapies, distinguishing between healthy and unhealthy cells, and recognition of previously unknown morphological traits [38,46,65]. The advantage of AI-based methodology lies in its independence from labelling, genetic alterations, or immunostaining, making it applicable to various scenarios that require the use of intact, live cells. Recently, AI technology has been employed to construct automated systems capable of high-throughput screening to reliably verify cells' identity and function throughout the entire production process. Furthermore, the integration of AI and robotic technology has led to the development of automated culture systems such as the CompacTSelecT (CTST) [113]. These advancements allow scientists to remotely implement protocols, facilitate the automatic maintenance and differentiation of iPSC into various cell types, and conduct highly efficient high-throughput screening using iPSC. This automation not only enhances the efficiency and consistency of cell culture workflows but also minimizes human error, leading to more reliable and reproducible results in stem cell research and applications.

For disease modelling, network-based screening using AI technologies is a powerful method for identifying molecules that address disrupted gene networks in human diseases, offering promising candidate molecules for in vivo validation [71,73] Furthermore, AI now extends its application beyond the generation of differentiated cells from iPSC. By integrating AI into the culture and maintenance of iPSC, it is possible to create complex tissue structures containing multiple cell types, essential for the development of organoids. These multicellular formations, illustrating three-dimensional tissue architecture, provide deeper insights into disease mechanisms compared to simpler cellular models. Recent studies highlight the significance of AI in refining the structure and functionality of organoids as models for disease [51,80]. Moreover, introduced in 2019, the DeepNEU platform, which integrates recurrent neural network (RNN), cognitive mapping (CM), and evolutionary systems, can exploit data from existing databases regarding essential genes and proteins that regulate and sustain

signalling pathways in hiPSC and hESC. This platform produces simulations of artificially induced pluripotent stem cell (aiPSC) that are consistent with the outcomes observed for actual iPSC [70]. This model has been successfully applied in generating aiPSC for infantile-onset Pompe disease (IOPD) and skeletal muscle, demonstrating high accuracy in replicating gene and protein expression and phenotypic characteristics [60]. These advancements have significant implications for the progression of disease research and clinical practices, highlighting the potential of AI-driven technologies in enhancing disease modelling, prototyping, and pathological prediction. This approach is particularly valuable for both common and rare diseases, offering a cost-effective alternative to traditional methods.

In drug discovery and development, AI is becoming increasingly integrated with iPSC technology, paving the way for novel, patient-specific assessment methods and the identification of new therapeutic agents for treating diseases. The synergy between AI and iPSC research extends beyond mere image analysis to the analysis of genomic data and pathological conditions [71,78,114]. These advancements open avenues for efficient selection, analysing the interactions and effectiveness of potential drug candidates through in silico virtual screening techniques, thereby revolutionizing drug discovery research with iPSC [115]. The initial research in this field was conducted by Eugene K. Lee et al. and Heylman Christopher in 2015. In these studies, iPSC-derived cardiomyocytes were used to evaluate drug-induced cardiotoxicity. Machine learning was employed to distinguish between normal and abnormal contractions [96] and membrane depolarization voltage following exposure to cardioactive drugs [95]. This approach showed an effective method in assessing the cardiotoxicity of various compounds, achieving an accuracy rate exceeding 80%. Subsequently, numerous studies have demonstrated the success of computational methods in predicting treatment responses [103] and in identifying internal changes post-treatment with certain substances [116] and pharmaceutical toxicity in iPSC and organoid models [94,104]. Currently, AI technology has been employed to develop high-throughput, label-free drug screening systems. AI-driven systems such as Deep-SeSMo and SSGraphCPI have demonstrated their potential in identifying new therapeutic agents to treat various diseases [117,118]. In 2020, Mahnaz Maddah et al. introduced PhenoTox, an 18-layer convolutional neural network system capable of detecting drug-induced structural changes in live hiPSC-hepatocytes and hiPSC-cardiomyocytes from bright-field microscopy images before these alterations are observable by humans. [119]. By 2022, Manuela Jaklin had developed the TeraTox system, a machine-learning algorithm with the ability to examine concentration-dependent cytotoxicity and changes in genetic expression induced by pharmaceutical compounds in hiPSC-derived embryoid bodies [100]. Particularly, during the COVID-19 pandemic in 2020, the DeepNEU was deployed for modelling lung aiPSC infection with SARS-CoV-2, facilitating the rapidly identification of antiviral therapeutic targets and opportunities for drug repurposing [61]. These advancements underscore AI's transformative role in drug development, highlighting its ability to deliver innovative and effective solutions in healthcare. The integration of AI and stem cell research not only expedites drug discovery processes but also paves the way for personalized medicine and targeted treatment strategies.

## Limitation and quality considerations in AI applications

The susceptibility of ML and DL algorithms to artifacts, along with the inherent biological complexity of cell-based assays, introduces numerous potential sources of noise and variability [120]. These challenges cover a broad spectrum of issues, including inconsistencies in cell plating, variations in cell growth and response, differential responses across plates, inter-plate variability, changes across different experimental runs, and edge effects—where conditions at the

edges of the plate are different from those at the center. In addition, clonal effects may add variability that complicates the analysis of results from cellular assays. Several strategies can be employed to address these challenges. Standardizing cell plating protocols, growth conditions, and assay procedures can markedly decrease variability. The adoption of automation and the use of high-throughput screening techniques can further standardize experimental workflows, reducing human error and inconsistency. Furthermore, the employment of advanced imaging and data analysis tools enables a more precise capture and analysis of biological responses, providing clearer insights into the data. Through these approaches, the reliability and accuracy of ML and DL applications in cell-based research can be significantly enhanced, paving the way for more robust scientific discoveries.

Moreover, the distinction between biological and technical replicates is another critical consideration. Algorithms must be trained and tested on data that accurately reflect both the biological variability and the technical reproducibility of the experiment. Failure to differentiate between these factors can lead to overfitting, where the model performs well on the training data but poorly on unseen data. To address this, it is essential to design experiments to ensure that the data used for training and testing the algorithms genuinely represent the diversity and complexity of biological systems. Furthermore, developing rigorous validation and cross-validation methods is important to verify the robustness and generalizability of AI models in cell-based research [121].

Another obstacle to integrating AI technologies into biological research is the difficulty in achieving reproducibility. This challenge stems from several factors, including the limited availability of the original computer codes and datasets, inadequate descriptions of the methodologies used, and the partial disclosure of findings [122]. Our analysis highlights that the code utilized was available in 29% of the cases, and only 3% of studies provided in best practice reports [29,97]. Additionally, the application of AI in biology is characterized by its diverse empirical methods, which, although may be successful in certain conditions and in specific laboratory environments, often do not have robust theoretical validation [123].

## Implications for future applications

AI holds substantial potential to revolutionize and accelerate the advancement of various aspects of iPSC technology, from the cell culturing process and drug discovery to disease modelling and the development of cellular therapies. Capable of extracting valuable insights from vast amounts of molecular and genomic data—tasks would be unfeasible for humans—AI could significantly advance iPSC research and development. Despite the potential benefits of AI in enhancing iPSC research, substantial technical hurdles must be overcome before its applications can be widely employed. Consequently, extensive research is required to address these barriers and fully harness the potential of AI in iPSC technology.

As advancements in AI technologies progress and the availability of high-quality data expands, the potential to refine and customize AI algorithms for application in regenerative medicine is expanding. Innovations in AI, computer vision, and robotics hold the promise of discovering new insights that could lead to transformative developments in iPSC technology. The integration of AI with other nanotechnology, genome editing, and 3D bioprinting could pave the way for breakthroughs in developing personalized regenerative therapies. By fostering interdisciplinary cooperation, ensuring the ethical development and application of these technologies, it is possible to fully leverage AI's potential to develop customized and effective regenerative treatments.

Finally, it is critical to establish robust benchmarking standards and criteria for the development and assessment of AI algorithms. These ground truth datasets, characterized by their

accurate labelling and broad acceptance, enable an unbiased evaluation of an algorithm's effectiveness. They serve as a cornerstone for comparing various models and methodologies and aiding in the discernment of the most efficient techniques for specific applications. Furthermore, standardized reporting protocols are crucial for significantly improving the quality of research in AI applications.

## Conclusions

Artificial intelligence technology, particularly ML and DL, offers substantial improvements over human capabilities in terms of accuracy, speed, and cost-efficiency in data analysis. These methods not only assist researchers but also pave the way for the transformation of decision-making processes in the biomedical field. AI technology has been significantly impacting every stage of stem cell studies, from laboratory research to clinical application. Its potential to revolutionize iPSC manufacturing lies in offering cost-effective, rapid, and robust screening methods for a large number of iPSC lines and their derivatives, essential for obtaining cells suitable for clinical use. Additionally, AI methods are increasingly being utilized in iPSC-based drug discovery, aiding drug efficacy, toxicity, and pharmacokinetics predictions.

The number of research studies in this area is constantly growing, pushing the boundaries of what AI can achieve in biomedical contexts. However, the implementation of these methods in clinical settings still requires substantial groundwork. A significant challenge lies in the need for extensive volumes of carefully curated, structured training data, which is crucial for generating meaningful results. Researchers must pay attention to factors that can influence the outcomes of AI applications, including the quality of data in the training set, the variability in sample size and unexpected events that algorithms may not predict.

### Limitations

Our study has certain limitations. Firstly, our research focuses on the application of AI in the context of iPSC therapy, we do not provide an in-depth analysis of the specific AI algorithms used in each referenced study. Second, there is inconsistency in reporting AI's accuracy, and, in some instances, it is missing. This inconsistency poses challenges in evaluating the efficacy of AI within iPSC therapy. Therefore, further detailed investigation and refinement of AI applications in iPSC therapy are necessary to fully understand and utilise their true therapeutic potential.

## Supporting information

**S1 Checklist. Preferred Reporting Items for Systematic reviews and Meta-Analyses extension for Scoping Reviews (PRISMA-ScR) checklist.**
(PDF)

**S1 File.**
(DOCX)

## Author Contributions

**Conceptualization:** Quan Duy Vo.

**Data curation:** Yukihiro Saito, Toshihiro Ida.

**Formal analysis:** Quan Duy Vo, Yukihiro Saito, Toshihiro Ida.

**Investigation:** Quan Duy Vo.

**Methodology:** Quan Duy Vo, Yukihiro Saito, Kazufumi Nakamura.

**Project administration:** Kazufumi Nakamura, Shinsuke Yuasa.

**Resources:** Quan Duy Vo.

**Supervision:** Kazufumi Nakamura, Shinsuke Yuasa.

**Validation:** Quan Duy Vo, Yukihiro Saito, Kazufumi Nakamura, Shinsuke Yuasa.

**Visualization:** Quan Duy Vo, Yukihiro Saito, Kazufumi Nakamura, Shinsuke Yuasa.

**Writing – original draft:** Quan Duy Vo, Yukihiro Saito, Toshihiro Ida.

**Writing – review & editing:** Quan Duy Vo, Yukihiro Saito, Kazufumi Nakamura, Shinsuke Yuasa.

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
