## [Decision Letter · Decision Letter 0]

8 Mar 2024

PONE-D-24-04149The utilization of Artificial Intelligence in induced pluripotent stem cell-based technology over 10-year period: a systematic scoping reviewPLOS ONE

Dear Dr. Vo,

Thank you for submitting your manuscript to PLOS ONE. After careful consideration, we feel that it has merit but does not fully meet PLOS ONE’s publication criteria as it currently stands. Therefore, we invite you to submit a revised version of the manuscript that addresses the points raised during the review process.

We look forward to receiving your revised manuscript.

Kind regards,

Li-Ping Liu

Academic Editor

PLOS ONE

Reviewers' comments:

Reviewer's Responses to Questions

**Comments to the Author**

1. Is the manuscript technically sound, and do the data support the conclusions?

Reviewer #1: Partly

Reviewer #2: Partly

Reviewer #3: No

Reviewer #4: Partly

Reviewer #5: Yes

Reviewer #6: Partly

Reviewer #7: Yes

2. Has the statistical analysis been performed appropriately and rigorously? 

Reviewer #1: N/A

Reviewer #2: N/A

Reviewer #3: N/A

Reviewer #4: N/A

Reviewer #5: N/A

Reviewer #6: Yes

Reviewer #7: Yes

3. Have the authors made all data underlying the findings in their manuscript fully available?

Reviewer #1: Yes

Reviewer #2: Yes

Reviewer #3: No

Reviewer #4: Yes

Reviewer #5: Yes

Reviewer #6: Yes

Reviewer #7: Yes

4. Is the manuscript presented in an intelligible fashion and written in standard English?

Reviewer #1: Yes

Reviewer #2: No

Reviewer #3: Yes

Reviewer #4: Yes

Reviewer #5: Yes

Reviewer #6: Yes

Reviewer #7: Yes

5. Review Comments to the Author

Reviewer #1: This manuscript by Vo et al. provides a comprehensive review of the application of artificial intelligence (AI) technologies, particularly machine learning and deep learning, in the processing of induced pluripotent stem cells (iPSCs). The authors have systematically reviewed 79 studies, highlighting the increasing trend of research in this area, with the United States being a leading contributor. The paper underscores the diverse applications of AI in iPSC technology, including cell type classification, disease-specific phenotype assessment in iPSC-derived cells, and facilitation of drug screening. The authors conclude by emphasizing the potential of AI technologies in advancing regenerative and personalized medicine, despite being in its early stages. They also highlight the challenges and opportunities in this field, suggesting that AI holds significant promise for enhancing our understanding of disease progression and development.

Although scientists have already written similar reviews on the interface of these two technologies in the last years (i.e. http://dx.doi.org/10.2174/1389201021666201007122524 or https://doi.org/10.3390%2Fbiomimetics8050442), this is a fast paced field and frequent literature reviews can help to understand progress. In this case the systematic nature of the review is also intriguing.

Before this work can be published, several aspects should be addressed (see major comments). Although it has a good structure, in general, the manuscript lacks sufficient depth for a scientific literature review. Key studies are rarely described in detail and several claims especially around human health and drug discovery improvements are made without clear connection to the literature.

Major comments:

1. Since the scope of the manuscript is the use of different AI-related technologies in the iPSC-field, I would expect a detailed and original figure giving a graphical summary of the frequency of the most commonly applied ML and DL algorithms. Which algorithms are used mostly? Is there a preference for certain algorithms in certain cell types? What are the main roles of the used algorithms (feature extraction, classification, regression, etc.)? The authors provide an excellent supplementary Table S1, which is a bit hidden, but is the core piece of this work. These results need to be more graphically visible. I would like to make the following proposition: First, merge Figure 2 and 3 into a new Figure 2 since these figures describe less interesting metadata. Second, use the data in Table S1 to create a new and insightful Figure 3 as described above.

2. Figures 4 and 5 have just been copied from two other publications and individually offer little to no new information. Also in this case I would encourage the authors to create a new Figure 4 by themselves which merges the information of the two current figures. In the new figure highlight specifically (and based on the literature) which kind if iPSC derived data (image features, molecular data, functional data) is generated or processed by which specific ML/DL algorithms and for what purpose (mentioned throughout the review). This will give a reader a clear understanding of the current literature. The current figures are just too general.

3. The authors make several strong claims about the use of AI and stem cell technology to improve human health. Although I agree that fundamental research has profited a lot, key clinical breakthroughs enabled by AI have yet to emerge (a new approved drug, stem cell therapy, approved disease models). For example, applying an AI algorithm to extract phenotypes from iPSC-cardiomyocytes is not yet a breakthrough for patients. Please back up these claims with literature or remove them. In general, go more in detail into key studies.

For example:

• “Recent advancement in artificial intelligence (AI) technologies, including machine learning, deep learning, and convolutional neural networks (CNN), has become fundamental in enhancing iPSC therapy.”

What is iPSC therapy (do you mean cell replacement?)? How has field profited exactly? Please go into key studies.

• “Recently, the application of Artificial Intelligence (AI) has significantly expanded, finding usage not just in regenerative medicine but across various domains in the medical field.”

Please be specific and go in detail into studies that describe this.

• “These findings highlighted that network-based screening using artificial intelligence technologies is a powerful method for identifying molecules that address disrupted gene networks in human diseases, offering promising therapy for in vivo validation.”

Therapy for in vivo validation does not make sense. It should be candidate molecules for in vivo validation.

• “These studies highlight the considerable potential of AI technology in the classification and assessment of iPSC-derived cells, significantly contributing to the advancement of disease modelling and drug development processes.”

How are drug development processes exactly positively impacted? Please be very specific and go more into individual studies.

• “This underscores AI's capability to analyse disease phenotypes, further emphasizing its role in advancing medical research and drug discovery.”

How exactly has this aided drug discovery? Which molecules in clinical phases are coming from an AI-aided iPSC system? Please be specific and mention the related work.

• “These developments highlight the significant impact of AI in drug development and disease research, emphasizing its capacity to provide innovative and efficient solutions in medical fields.”

• “The advancement of AI-driven systems such as Deep-SeSMo and SSGraphCPI demonstrates their potential in identifying new therapeutic agents to treat various diseases.”

That might be true, but please give detailed examples from the actual studies.

• “[…] has become fundamental in enhancing iPSC therapy.”

4. The above mentioned sections should be removed, toned down or ideally backed up with detailed literature information. The preferred option would be to add a new dedicated section on how iPSC-derived and ML/DL processed data has contributed to the drug development process. This would be in line with the existing sections on Cell culture, Disease models, Genetic analysis, Functional monitoring etc.

5. A very big issue in the field of ML/DL in biology is reproducibility. In would be important to add another figure or table describing which percentage of papers make code and data available and how well the ML methods and workflows are described. Do papers describe whether hyperparameter tuning was performed? How many papers follow best practice guidelines? For example the DOME (data, optimization, model, and evaluation) recommendations (https://doi.org/10.1038/s41592-021-01205-4) or similar best practices.

Minor comments:

1. Remove judgemental adjectives such as “impressive” or “notable success”.

2. The Discussion part is in the middle of the manuscript and should be moved more to the end.

3. Below Figure 4 (I believe) the section header “Quality control” is missing. At least I have the impression that this section is on quality control and has no header.

4. In the section “Cell function monitoring” mostly Calcium assays are described. However, microelectrode arrays (MEAs) are also routinely used to measure muscle or neuron electrophysiological activity. This kind of data is also often analyzed by ML. Please include these studies as well. For example: https://doi.org/10.1038/s41598-022-05697-8 or https://doi.org/10.3389%2Ffnins.2021.647877.

5. In the Discussion or Conclusion section go into the fact that ML/DL algorithms (like any technique) are sensitive to artifacts that can sometimes be avoided or at least be controlled for. For example: Plate position (edge effects), clonal effects, biological versus technical replicates (training and testing data from the same replicate?), need for good benchmarking and ground truth data sets.

Reviewer #2: In this manuscript, the authors do a systematic search for all studies that involve AI and induced pluripotent stem cell-based research. This work is timely to address the increasingly popular field of AI and its application in stem cell research.

Major concerns

1. There is a large gap between the systematic scoping review in the results and the discussion. Consider discussing more regarding the findings of the scoping review, such as the trends over the years – of complexity of the datasets studied (simple purchased iPSCS compared to large group of iPSCs from clinical studies), of applications of the findings (image recognition of iPSC clonal expansion compared to large scale disease models), and the complexity of the machine learning algorithms used (Simple random forest/SVMs compared to deep neural networks).

2. The discussion is extremely text heavy. Summarise the many examples within each section using tables, highlighting the AI models used, their applications, and improvements over previous iterations of the same application.

3. What were the main inclusion criteria that the records failed to meet that resulted in the n=432 records being excluded to n=97? That is a huge drop that deserves to be addressed. Authors will need to describe in detail the selection process.

4. Describe how the current studies may potentially benefit the use of AI and machine learning in tissue organoid culture and applications.

Minor comments

1. Several abbreviations were repeated.

Reviewer #3: The authors reviewed articles to highlight the advancement of combination of AI and iPS, and showcased the future promising advancement of this field.

The concept sounds interesting. Overall, this review article is written well.

Major Comment:

Please declare the publishers’ permission for reuse of the figures (Fig 4, 5, and 6).

Minor Comments:

There are so many times of abbreviation definitions in the main text body like AI, ML, DL, CNN, iPSCs, AUC, ALS. Please define one time at the first time in each highest section i.e. Abstract, Introduction, Result and Discussion. However in abstract, first AI and iPSC appears without full spelling i.e. artificial intelligence (AI). Please unify the definitions for iPSCs or iPSC, CNNs or CNN. I found similar miss writings. Please read once more carefully.

At page 8, line 4, capitalization of “Artificial Intelligence” (AI) is not necessary.

Reviewer #4: This is a very interesting review manuscript, the authors summarized the applications of AI algorithms in iPSCs based studies. I have a few comments for the authors to address.

1. In introductions, it reads: "Recent advancement in artificial intelligence (AI) technologies, including machine learning, deep learning, and convolutional neural networks (CNN), has become fundamental in enhancing iPSC therapy. " CNN is a deep leaning algorithm, should not be listed separately.

2. In Methods, for the Inclusion and Exclusion Criteria, I am not sure about if the full text not available should be an exclusion criteria. How many studies were excluded because their full text was not available for this review?

3. In Disease model section, what is deep machine learning? was it a typo?

4. For the future applications section, it should outline the new directions and provide perspectives of this field. Current version is very repetitive to the applications sections. This part needs to be re written.

5. Limitations of the AI technologies in stem cell research areas should be discussed as well.

Reviewer #5: The manuscript is very interesting and made carefully. Data originating from induced pluripotent stem cells are nowadays extensively studied and artificialmn intelligence methods are very relevant for such research efforts.

Minor corrections:

p. 8, paragraph 3, row 1: Change "employs artificial neural networks" to "employs, among others, artificial neural networks". Viz., deep learning also covers other machine learning methods than artificial neural networks, although these are the best known and perhaps most used in deeep learning.

p. 13, row 2: Change "ROC (Receiver Operating Chracteristics) of 0.93" to "AUC (area under the Receiver Operating Chracteristics, ROC, curve) of 0.93"

Reviewer #6: In this paper, they conducted a systematic review on the utilization of AI in iPSC technology. A total of 79 papers were included and analyzed, yielding interesting results. This research is considered valuable for advancing the use of AI with iPSCs in the future.

Overall, there is inconsistency in the explanations of terms such as machine learning, deep learning, neural networks, and AI. Also, there seems to be confusion about whether each term is a higher-level concept, a lower-level concept, or a comparative term. Please review the manuscript again to ensure consistency of terms, and to avoid contradictions.

＜Specific comments＞

1. Introduction section (page 4). They state, "The Convolutional Neural Network (CNN), a supervised learning method, has significantly enhanced the outcomes of image recognition studies." However, while CNN is commonly associated with supervised learning, it is also utilized in unsupervised learning and reinforcement learning. Therefore, please revise the manuscript accordingly.

2. In Table S1, they have divided the overall algorithms into deep learning and machine learning, but this may cause confusion. Deep learning is a subset of machine learning technique, so deep learning and machine learning should not be compared as equal terms. Please correct this for clarity.

3. In Figure 4, they argue that deep learning is a part of AI. In this case, AI seems to refer to general AI. On the other hand, the main focus of this paper is on specialized AI for tasks such as image analysis. To avoid confusion, please clarify whether the discussion pertains to general AI or specialized AI

4. On page 10, it is unclear what "The integration of machine learning, deep learning, and convolutional neural networks (CNN)" is explaining. Please provide specific descriptions.

5. The text length of discussion section is much longer than that of results section, making it unclear what is the findings of the research, and the discussion based on findings. For example, they discuss detailed contents of the papers listed in Table S1 within the discussion section. However, some of these should be included in the results section. The content of the discussion and the results should be restructured.

Reviewer #7: Nice survey and analysis of the AI/ML applications in stem cell research.

It will be much better to include analysis on what algorithms of AI/ML are used, % among the qualified studies, while keeping in mind that the algorithms are continuously being developed.

6. PLOS authors have the option to publish the peer review history of their article (what does this mean?). If published, this will include your full peer review and any attached files.

Reviewer #1: **Yes: **Johannes H Wilbertz

Reviewer #2: No

Reviewer #3: No

Reviewer #4: No

Reviewer #5: No

Reviewer #6: No

Reviewer #7: No

---

## [Author Response · Author response to Decision Letter 0]

15 Mar 2024

Thank you for your comprehensive feedback. Your insights are invaluable in enhancing the quality of our manuscript. Here are our responses to the reviewers’ comments.

Reviewer 1:

1. Since the scope of the manuscript is the use of different AI-related technologies in the iPSC-field, I would expect a detailed and original figure giving a graphical summary of the frequency of the most commonly applied ML and DL algorithms. Which algorithms are used mostly? Is there a preference for certain algorithms in certain cell types? What are the main roles of the used algorithms (feature extraction, classification, regression, etc.)? The authors provide an excellent supplementary Table S1, which is a bit hidden, but is the core piece of this work. These results need to be more graphically visible. I would like to make the following proposition: First, merge Figure 2 and 3 into a new Figure 2 since these figures describe less interesting metadata. Second, use the data in Table S1 to create a new and insightful Figure 3 as described above.

Response: We have created Fig 3 that combines the information about the frequency of ML and DL algorithms and the roles of the used algorithms in each research aim. Also, we have created Table 1 – 5 to provide further information about the use of AI algorithms in each study.

2. Figures 4 and 5 have just been copied from two other publications and individually offer little to no new information. Also in this case I would encourage the authors to create a new Figure 4 by themselves which merges the information of the two current figures. In the new figure highlight specifically (and based on the literature) which kind if iPSC derived data (image features, molecular data, functional data) is generated or processed by which specific ML/DL algorithms and for what purpose (mentioned throughout the review). This will give a reader a clear understanding of the current literature. The current figures are just too general.

Response: We have created a new Figure 1 to replace Figure 4 and Figure 5.

3. The authors make several strong claims about the use of AI and stem cell technology to improve human health. Although I agree that fundamental research has profited a lot, key clinical breakthroughs enabled by AI have yet to emerge (a new approved drug, stem cell therapy, approved disease models). For example, applying an AI algorithm to extract phenotypes from iPSC-cardiomyocytes is not yet a breakthrough for patients. Please back up these claims with literature or remove them. In general, go more in detail into key studies.

- “Recent advancement in artificial intelligence (AI) technologies, including machine learning, deep learning, and convolutional neural networks (CNN), has become fundamental in enhancing iPSC therapy.”

What is iPSC therapy (do you mean cell replacement?)? How has field profited exactly? Please go into key studies. 

Response: In this case, we want to mention regenerative therapy. We have clarified the information and provided the literature reference.

- “Recently, the application of Artificial Intelligence (AI) has significantly expanded, finding usage not just in regenerative medicine but across various domains in the medical field.”

Please be specific and go into detail studies that describe this.

Response: We have removed this statement.

- “These findings highlighted that network-based screening using artificial intelligence technologies is a powerful method for identifying molecules that address disrupted gene networks in human diseases, offering promising therapy for in vivo validation.”

Therapy for in vivo validation does not make sense. It should be candidate molecules for in vivo validation.

Response: We have corrected the sentence.

- “These studies highlight the considerable potential of AI technology in the classification and assessment of iPSC-derived cells, significantly contributing to the advancement of disease modelling and drug development processes.”

 How are drug development processes exactly positively impacted? Please be very specific and go more into individual studies.

- “This underscores AI's capability to analyse disease phenotypes, further emphasizing its role in advancing medical research and drug discovery.”

How exactly has this aided drug discovery? Which molecules in clinical phases are coming from an AI-aided iPSC system? Please be specific and mention the related work.

Response: We have removed these sentences and provided further information in the Discussion section.

- “These developments highlight the significant impact of AI in drug development and disease research, emphasizing its capacity to provide innovative and efficient solutions in medical fields.”

- “The advancement of AI-driven systems such as Deep-SeSMo and SSGraphCPI demonstrates their potential in identifying new therapeutic agents to treat various diseases.”

That might be true, but please give detailed examples from the actual studies.

“[…] has become fundamental in enhancing iPSC therapy.”

The above-mentioned sections should be removed, toned down or ideally backed up with detailed literature information. The preferred option would be to add a new dedicated section on how iPSC-derived and ML/DL processed data has contributed to the drug development process. This would be in line with the existing sections on Cell culture, Disease models, Genetic analysis, Functional monitoring etc.

Response: We have rewritten the drug development part in Discussion section and modified the words and provided the literature references for these sentences.

4. A very big issue in the field of ML/DL in biology is reproducibility. In would be important to add another figure or table describing which percentage of papers make code and data available and how well the ML methods and workflows are described. Do papers describe whether hyperparameter tuning was performed? How many papers follow best practice guidelines? For example the DOME (data, optimization, model, and evaluation) recommendations (https://doi.org/10.1038/s41592-021-01205-4) or similar best practices.

Response: We have provided the information about the quality of studies in Quality of included research section, and also provided detail information in Table S2.

5. Remove judgmental adjectives such as “impressive” or “notable success”.

Response: We have removed the judgmental adjectives.

6. The Discussion part is in the middle of the manuscript and should be moved more to the end.

Response: We have reorganized the manuscript for more comprehension.

7. Below Figure 4 (I believe) the section header “Quality control” is missing. At least I have the impression that this section is on quality control and has no header.

Response: We have added the label for this section.

8. In the section “Cell function monitoring” mostly Calcium assays are described. However, microelectrode arrays (MEAs) are also routinely used to measure muscle or neuron electrophysiological activity. This kind of data is also often analyzed by ML. Please include these studies as well. For example: https://doi.org/10.1038/s41598-022-05697-8 or https://doi.org/10.3389%2Ffnins.2021.647877. 

Response: We have added the information about MEA in the Discussion section.

9. In the Discussion or Conclusion section go into the fact that ML/DL algorithms (like any technique) are sensitive to artifacts that can sometimes be avoided or at least be controlled for. For example: Plate position (edge effects), clonal effects, biological versus technical replicates (training and testing data from the same replicate?), need for good benchmarking and ground truth data sets.

Response: We have added the information in Quality considerations in AI research.

Reviewer 2: 

1. There is a large gap between the systematic scoping review in the results and the discussion. Consider discussing more regarding the findings of the scoping review, such as the trends over the years – of complexity of the datasets studied (simple purchased iPSCS compared to large group of iPSCs from clinical studies), of applications of the findings (image recognition of iPSC clonal expansion compared to large scale disease models), and the complexity of the machine learning algorithms used (Simple random forest/SVMs compared to deep neural networks).

Response: we have discussed the Evolution of AI in iPSC Research in the Discussion section.

2. The discussion is extremely text heavy. Summarise the many examples within each section using tables, highlighting the AI models used, their applications, and improvements over previous iterations of the same application.

Response: We have rewritten the Result and discussion part in order to reduce the text heavy.

3. What were the main inclusion criteria that the records failed to meet that resulted in the n=432 records being excluded to n=97? That is a huge drop that deserves to be addressed. Authors will need to describe in detail the selection process.

Response: We have incorporated detailed information regarding the selection process in Figure 2. The primary criteria for exclusion encompass documents that are not original research, including reviews, book chapters, letters to the editor, conference posters, and studies that did not utilize iPSC or AI technology.

4. Describe how the current studies may potentially benefit the use of AI and machine learning in tissue organoid culture and applications.

Response: we have added the inforation about application of AI and machine learning in organoid technology in Discussion section.

5. Several abbreviations were repeated.

Response: we have checked and excluded repeated abbreviations.

Reviewer 3:

1. Please declare the publishers’ permission for reuse of the figures (Fig 4, 5, and 6).

Response: 

- We have replaced the Fig 4, Fig 5 and Fig 6 and replaced them by original Figures and tables. 

2. There are so many times of abbreviation definitions in the main text body like AI, ML, DL, CNN, iPSCs, AUC, ALS. Please define one time at the first time in each highest section i.e. Abstract, Introduction, Result and Discussion. However in abstract, first AI and iPSC appears without full spelling i.e. artificial intelligence (AI). Please unify the definitions for iPSCs or iPSC, CNNs or CNN. I found similar miss writings. Please read once more carefully.

At page 8, line 4, capitalization of “Artificial Intelligence” (AI) is not necessary.

Response: we have corrected the errors about abbreviations. We changed iPSCs to iPSC, CNNs to CNN

Reviewer 4:

1. In introductions, it reads: "Recent advancement in artificial intelligence (AI) technologies, including machine learning, deep learning, and convolutional neural networks (CNN), has become fundamental in enhancing iPSC therapy. " CNN is a deep leaning algorithm, should not be listed separately.

Response: We have corrected this sentence.

2. In Methods, for the Inclusion and Exclusion Criteria, I am not sure about if the full text not available an exclusion should be criteria. How many studies were excluded because their full text was not available for this review?

Response: In our analysis, only two studies were excluded because they were posters presented at conferences, from which we were unable to obtain the full text manuscripts.

3. In Disease model section, what is deep machine learning? was it a typo?

Response: Actually, it is deep learning. We have corrected the typo.

4. For the future applications section, it should outline the new directions and provide perspectives of this field. Current version is very repetitive to the applications sections. This part needs to be re written.

Response: We have rewritten this Future applications section

5. Limitations of the AI technologies in stem cell research areas should be discussed as well.

Response: We have added Limitations of the AI technologies section

Reviewer 5: 

- p. 8, paragraph 3, row 1: Change "employs artificial neural networks" to "employs, among others, -artificial neural networks". Viz., deep learning also covers other machine learning methods than artificial neural networks, although these are the best known and perhaps most used in deeep learning.

- p. 13, row 2: Change "ROC (Receiver Operating Chracteristics) of 0.93" to "AUC (area under the Receiver Operating Chracteristics, ROC, curve) of 0.93"

Response: We have changed the word accordingly.

Reviewer 6: 

1. Overall, there is inconsistency in the explanations of terms such as machine learning, deep learning, neural networks, and AI. Also, there seems to be confusion about whether each term is a higher-level concept, a lower-level concept, or a comparative term. Please review the manuscript again to ensure consistency of terms, and to avoid contradictions.

Response: we have reviewed the manuscript and corrected the terms’ definitions.

2. Introduction section (page 4). They state, "The Convolutional Neural Network (CNN), a supervised learning method, has significantly enhanced the outcomes of image recognition studies." However, while CNN is commonly associated with supervised learning, it is also utilized in unsupervised learning and reinforcement learning. Therefore, please revise the manuscript accordingly.

Response: we have corrected the sentence and removed “supervised learning method” statement.

3. In Table S1, they have divided the overall algorithms into deep learning and machine learning, but this may cause confusion. Deep learning is a subset of machine learning technique, so deep learning and machine learning should not be compared as equal terms. Please correct this for clarity.

Response: we have removed the comparison between deep learning and machine learning in Table S1.

4. In Figure 4, they argue that deep learning is a part of AI. In this case, AI seems to refer to general AI. On the other hand, the main focus of this paper is on specialized AI for tasks such as image analysis. To avoid confusion, please clarify whether the discussion pertains to general AI or specialized AI.

Response: We have changed the Figure 4 for the new original Figure, and also clarify the discussion pertains to general AI.

5. On page 10, it is unclear what "The integration of machine learning, deep learning, and convolutional neural networks (CNN)" is explaining. Please provide specific descriptions.

Response: we have corrected the sentence to be more clarified.

6. The text length of discussion section is much longer than that of results section, making it unclear what is the findings of the research, and the discussion based on findings. For example, they discuss detailed contents of the papers listed in Table S1 within the discussion section. However, some of these should be included in the results section. The content of the discussion and the results should be restructured.

Response: We have rewritten the result and discussion section. Also, we have reduced the number of word and replaced by tables and images. 

Reviewer 7: 

It will be much better to include analysis on what algorithms of AI/ML are used, % among the qualified studies, while keeping in mind that the algorithms are continuously being developed.

Response: we have added the new Figure 4 to clarify this information. Also, we have added a discussion about this problem in Result section.

---

## [Decision Letter · Decision Letter 1]

2 Apr 2024

PONE-D-24-04149R1The utilization of artificial intelligence in induced pluripotent stem cell-based technology over 10-year period: A Systematic Scoping ReviewPLOS ONE

Dear Dr. Vo,

Thank you for submitting your manuscript to PLOS ONE. After careful consideration, we feel that it has merit but does not fully meet PLOS ONE’s publication criteria as it currently stands. Therefore, we invite you to submit a revised version of the manuscript that addresses the points raised during the review process.

We look forward to receiving your revised manuscript.

Kind regards,

Li-Ping Liu

Academic Editor

PLOS ONE

Journal Requirements:

Reviewers' comments:

Reviewer's Responses to Questions

**Comments to the Author**

1. If the authors have adequately addressed your comments raised in a previous round of review and you feel that this manuscript is now acceptable for publication, you may indicate that here to bypass the “Comments to the Author” section, enter your conflict of interest statement in the “Confidential to Editor” section, and submit your "Accept" recommendation.

Reviewer #1: All comments have been addressed

Reviewer #2: All comments have been addressed

2. Is the manuscript technically sound, and do the data support the conclusions?

Reviewer #1: Yes

Reviewer #2: Yes

3. Has the statistical analysis been performed appropriately and rigorously? 

Reviewer #1: Yes

Reviewer #2: N/A

4. Have the authors made all data underlying the findings in their manuscript fully available?

Reviewer #1: Yes

Reviewer #2: Yes

5. Is the manuscript presented in an intelligible fashion and written in standard English?

Reviewer #1: Yes

Reviewer #2: Yes

6. Review Comments to the Author

Reviewer #1: I would like to thank the authors for their extensive work on the revised manuscript. Limitations of the included studies are now better explained and the new figures and additional tables convey the key information extracted from the literature better.

Please still carefully revise the text. I found a couple of spelling mistakes or missing words (but there might be more):

- I would like to propose a change of the main title: The word “utilization” is rarely used. A simple synonym such as “use” would be better.

- In the sub-section header “Main AI algorithm employed in iPSC research” the word “algorithm” should be in its plural form “algorithms”.

- The sub-section header “Applications of AI in iPSC” should be “Applications of AI in iPSC research” or something similar.

- “Fig 1. The application of AI technology in iPSC field” should be “Fig 1. The application of AI technology in the iPSC field.” (Punctuation and the word “the” was missing.

- The authors often use “iPSC” as a synonym for the word “technology”, but iPSC essentially just means “cells”. For example: In the sentence “AI has been instrumental in advancing iPSC development […]” the word “technology” should be added: “AI has been instrumental in advancing iPSC technology development […]”. Please make sure to check this throughout the manuscript.

Reviewer #2: The authors have made substantial changes to the manuscript and have answered all my queries. I have no further questions.

7. PLOS authors have the option to publish the peer review history of their article (what does this mean?). If published, this will include your full peer review and any attached files.

Reviewer #1: **Yes: **Johannes Wilbertz

Reviewer #2: No

---

## [Author Response · Author response to Decision Letter 1]

4 Apr 2024

Reviewer 1:

- I would like to propose a change of the main title: The word “utilization” is rarely used. A simple synonym such as “use” would be better.

- In the sub-section header “Main AI algorithm employed in iPSC research” the word “algorithm” should be in its plural form “algorithms”.

- The sub-section header “Applications of AI in iPSC” should be “Applications of AI in iPSC research” or something similar.

- “Fig 1. The application of AI technology in iPSC field” should be “Fig 1. The application of AI technology in the iPSC field.” (Punctuation and the word “the” was missing.

- The authors often use “iPSC” as a synonym for the word “technology”, but iPSC essentially just means “cells”. For example: In the sentence “AI has been instrumental in advancing iPSC development […]” the word “technology” should be added: “AI has been instrumental in advancing iPSC technology development […]”. Please make sure to check this throughout the manuscript.

Response: We have changed our manuscript following the reviewer’s suggestion and carefully check the grammar mistakes in our manuscript.

---

## [Editor Report · Decision Letter 2]

9 Apr 2024

The use of artificial intelligence in induced pluripotent stem cell-based technology over 10-year period: A Systematic Scoping Review

PONE-D-24-04149R2

Dear Dr. Vo,

We’re pleased to inform you that your manuscript has been judged scientifically suitable for publication and will be formally accepted for publication once it meets all outstanding technical requirements.

Kind regards,

Li-Ping Liu

Academic Editor

PLOS ONE
---

## [Editor Report · Acceptance letter]

9 May 2024

PONE-D-24-04149R2 

PLOS ONE

Dear Dr. Vo, 

I'm pleased to inform you that your manuscript has been deemed suitable for publication in PLOS ONE. Congratulations! Your manuscript is now being handed over to our production team.

Kind regards, 

on behalf of

Dr. Li-Ping Liu 

Academic Editor

PLOS ONE